# Distinct structures of interannual variations in stratosphere-to-troposphere ozone transport induced by the Tibetan Plateau thermal forcing

Qingjian Yang[1], Tianliang Zhao[1], Yongqing Bai[2], Kai Meng[3], Yuehan Luo[1], Zhijie Tian[4], Xiaoyun Sun[5], Weikang Fu[6], Kai Yang[1], Jun Hu[7]

[1]Key Laboratory for Aerosol-Cloud-Precipitation of China Meteorological Administration, Nanjing University of Information Science &Technology, Nanjing 210044, China
[2]Hubei Key Laboratory for Heavy Rain Monitoring and Warning Research, Institute of Heavy Rain, China Meteorological Administration, Wuhan 430205, China
[3]Key Laboratory of Meteorology and Ecological Environment of Hebei Province, Hebei Provincial Institute of Meteorological Sciences, Shijiazhuang 050021, China
[4]China Institute for Radiation Protection, Taiyuan 030006, China
[5]Anhui Province Key Laboratory of Atmospheric Science and Satellite Remote Sensing, Anhui Institute of Meteorological Sciences, Hefei 230031, China
[6]Public Meteorological Service Center, China Meteorological Administration, Beijing 100081, China
[7]Fujian Academy of Environmental Sciences, Fuzhou 350011, China

*Correspondence to*: Tianliang Zhao (tlzhao@nuist.edu.cn)

**Abstract.** Stratosphere-to-troposphere transport (STT) is an essential natural source of tropospheric ozone. Focusing on the STT variations in late spring and early summer with frequent ozone STT over the Asian region, this study investigated the Tibetan Plateau (TP) thermal effect on the interannual STT variations with the meteorological data over 1980–2014. The distinct structures of interannual STT anomalies with the key areas of TP thermal forcing are identified through the 35-year climatological statistics. Positive anomalies of thermal forcing over the central and eastern TP exert opposing impacts on the increasing and decreasing ozone STT respectively along northern and southern branches of the westerly jet around the TP. Such the stronger TP thermal forcing induces anticyclonic anomalies in the upper troposphere over the TP and the surroundings, which strengthens and attenuates the northern and southern branches of the westerly jet, intensifying and weakening the westerly trough for more (less) tropopause folds of ozone STT over East Asian region. Furthermore, the positive anomalies of thermal forcing over the western TP are related to the western enhancing and eastern declining STT over the Asian region. This study reveals the distinct structures of interannual variations in ozone STT induced by the TP thermal forcing, providing a new prospect for the TP effect on atmospheric environment.

## 1 Introduction

As one of the most important chemicals in the atmosphere, ozone has globally captured considerable attention and sparked extensive scientific investigations. The high ozone layer in the stratosphere absorbs most of the harmful UV radiation,

protecting the biosphere on the earth's surface. Conversely, ozone in the troposphere acts as a potent oxidant with adverse effects on human health (Nuvolone et al., 2018), crop growth (Feng et al., 2015), and ecosystems (Yue et al., 2017). Despite its abundance in the troposphere being much less than that in the stratosphere (10% vs. 90%), tropospheric ozone can influence global radiation balance and then the multiscale climate patterns of global, hemispheric, and regional circulations in the atmosphere such as high-latitude warming, Hadley circulation, and East Asian summer monsoon system (Chen et al., 2007; Li et al., 2018) as an effective greenhouse gas (Schoeberl and Hartmann, 1991).

Although it is well-established that the photochemical oxidation of volatile organic compounds and nitrogen oxides primarily produces tropospheric ozone (Sillman, 2003), stratosphere-to-troposphere transport (STT) was proved to be an essential natural source influencing ozone change in the troposphere (Meng et al., 2022; Zhao et al., 2021). STT can transport ozone-enriched air from the stratosphere downward to the troposphere, exerting an important impact on temporal-spatial variations in tropospheric ozone (Ni et al., 2019; Oltmans et al., 2004). Thus, exploring the factors influencing the changes in ozone STT is of great importance.

The STT occurrences or stratospheric intrusions are highly associated with the upper-tropospheric westerly jet in the middle latitudes (Ding and Wang, 2006; Hsu et al., 2005; Lee et al., 2024). The descending pathway of STT is attributed to the tilted isentropic surfaces along the strong westerlies in the midlatitudes (Akritidis et al., 2016; Škerlak et al., 2014; Sprenger & Wernli, 2003). The upper-level jet can drive isentropic mixing near the jet axis for tropopause folds, initiating downward transport of stratospheric ozone with influence on the ozone in the troposphere (Lee et al., 2024).

Climatologically, the westerly jet is located near the tropopause between 200 hPa and 300 hPa, with the southern branch extending northeastward from the southern side of the Tibetan Plateau (TP) to the Northwest Pacific and the northern branch stretching southeastward from the 40°–65°N region north of the TP to Northwest Pacific. The two branches converge over the northwestern Pacific, forming the strongest jet stream in the world (Cressman, 1981). The TP terrain effect plays a crucial role in shaping the southern and northern branches of westerly jet (Li and Liu, 2015; Liu et al., 2020). During spring and summer, the thermal forcing of the TP induces the Asian summer monsoon anticyclonic circulation in the upper troposphere and lower stratosphere (UTLS) (Wu et al., 2007), intensifying the northern branch westerly jet northward, and inhibiting the southern branch westerly jet, which may even disappear by late June (Duan and Wu, 2005; Luo and Yanai, 1983). The increase in spring snow cover with negative anomalies of TP thermal forcing over the TP can strengthen the southern branch and weaken the northern branch of the westerly jet (Dong et al., 1997). The north-south shift of the westerly jet is closely linked to the seasonal variations of the TP thermal forcing, especially during the spring-summer seasonal transition period (Li and Liu, 2015). However, it is poorly understood that the TP thermal forcing anomalies affect the changes of ozone STT along the northern and southern branches of the westerly jet.

The TP serves as a large heat source in boreal summer because high surface sensible heating could cause cyclonic circulation anomalies in the lower troposphere and could trigger vertical motion with notable latent heat release from cloud-precipitation in the middle-upper troposphere over the TP (Luo and Yanai, 1984; Duan and Wu, 2005; Wu et al., 2007). The TP thermal forcing refers to the heating intensity in the atmosphere over the TP, resulting from the atmospheric diabatic process of

surface sensible heat, latent heat, and radiation heat. As the roof of the world with a large climatic external forcing, the TP can induce westerly jet change and further influence the weather (Wu et al., 2007; Xu et al., 2010, 2022; Tan et al., 2023) and atmospheric environment in the downstream regions. The anomalous westerly jet induced by the TP significantly affects dust aerosol transport, precipitation, and drought in China (Liu et al., 2020). The thermal forcing change over the TP significantly modulates the interannual variations in ozone pollution with a dipole pattern over Eastern China (Yang et al., 2024). However, the relationship between the TP terrain effect and ozone STT over China in the interannual variations remains unclear. Given that the thermal forcing of the TP significantly modulates the intensity and position of the westerly jet, the objective of this study is to characterize the thermal effect of the TP on STT of ozone change over China with comprehending the TP's effect on atmospheric environment change.

Based on the data of STT frequency and ozone STT product over 1980–2014 combined with the ERA5 data of reanalysis meteorology, the impact of the TP thermal forcing on interannual variations in ozone STT over China was investigated by using the statistical methods of Singular Value Decomposition (SVD), composite analysis, and correlation analysis. Following Sect. 1 Introduction, the data and methodology are presented in Sect. 2. The impacts of TP thermal effect on ozone STT are explored in Sect. 3. In Sect. 4, the physical mechanisms are investigated. In Sect. 5, we provide the conclusions of this study. The aim of this study is to extend the environmental influence of the TP terrain forcing on STT of ozone over downstream regions for a comprehensive understanding of environmental changes in China.

## 2. Data and Methodology

### 2.1 Data

The monthly STT frequency data from 1980 to 2014 were provided by the atmospheric dynamics group at ETH Zürich (http://eraiclim.ethz.ch/prot/ste.html). The detection algorithm for STT events initiates with a comprehensive grid of trajectories launched every 24 hours. Those trajectories intersecting the tropopause within the first day are identified and then tracked for four days, both forwards and backwards in time. In order to eliminate the transient exchange events, the methodology requires that each trajectory remains on either side of the tropopause for a minimum of 48 hours. For the detailed introduction of the computational methods, please refer to the study by Sprenger and Wernli, (2003).

The monthly data of ozone STT mass flux ($kg\ km^{-2}\ month^{-1}$) from 1980 to 2014 was further calculated based on the STT frequency (Škerlak et al., 2014). Each trajectory represents a fixed amount of air mass flux determined by the spacing of the starting grid. Based on the air mass flux, the ozone STT flux is calculated accordingly with the molecular weights of the dry air and ozone as well as the ozone concentrations at the tropopause. A thorough guide on the calculation method of the ozone STT mass flux is provided in the study of Škerlak et al. (2014).

The monthly tropopause fold frequency data from 1980 to 2014 were also derived from ETH Zürich (http://eraiclim.ethz.ch/prot/folds.html). The identification of tropopause folds consists of a three-dimensional subdivision of the atmosphere into stratospheric and tropospheric parts and a subsequent examination of multiple tropopause crossings in

vertical profiles of the analyses. The detailed calculation method can be referred to Tyrlis et al. (2014) and Škerlak et al.
(2015).

The monthly reanalysis data of meteorology with a horizontal resolution of 0.25 $°×$0.25 $°$ and 32 vertical pressure layers up to 10 hPa were derived from the ERA5 of the European Centre for Medium-Range Weather Forecasts (ECMWF) (Hersbach et al., 2020) reanalysis dataset (https://www.ecmwf.int/en/forecasts/datasets/reanalysis-datasets/era5/), including air temperature (T), potential vorticity (PV), relative humidity (RH), latitudinal and meridional components (U, V) of wind, and
vertical velocity ($\omega$).

**2.2 The heat source over the TP**

The atmospheric heat source (Q1) is a physical quantity that reflects the heat balance of an air column. For a given area, the Q1 is defined as the heat gained (lost) by the air column over a period of time, which is a combination of sensible heat, latent heat of condensation, and atmospheric radiation. Following the inverse algorithm of Yanai et al. (1973, 1992), the
atmospheric heat source (Q1) was calculated based on the ERA5 reanalysis data:

$$Q_1 = c_p \left[ \frac{\partial T}{\partial t} + V \cdot \nabla T + \left( \frac{p}{p_0} \right)^k \omega \frac{\partial \theta}{\partial p} \right] \tag{1}$$

where T and $\theta$ are air temperature and potential temperature, respectively; k = R/Cp, where R and Cp are the dry air gas constant and specific heat at constant pressure, respectively; $P_0$ is 1000 hPa.

The vertical integral value of Q1 in an atmospheric column can be expressed as:

$$(Q_1) = \frac{1}{g} \int_{P_t}^{P_s} Q_1 dp \tag{2}$$

where $P_s$ and $P_t$ denote the surface pressure and atmospheric top pressure (200 hPa), respectively, and g is gravitational acceleration. The vertical integral of Q1 over the TP area of 78–103 $°$E and 28–38 $°$N, covering most of the region with an altitude >3000 m (Xu et al., 2016), represents the thermal forcing of the TP.

**2.3 Singular Value Decomposition (SVD)**

SVD is a powerful mathematical technique to identify the spatio-temporal relationships between two variable fields. Since the first application of SVD in meteorological analysis by Prohaska (1976), SVD has been widely applied in the studies of environment, climate, and ecology (Sein et al., 2024; Wang et al., 2017; Zhou et al., 2022). The matrix decomposition of SVD can obtain the spatial modes and their corresponding temporal coefficients from the correlation fields of paired variable domains. The correlations between the variations of one domain and the temporal coefficient of the other are defined as the
heterogeneous correlation coefficients, reflecting the inter-field correlation distribution. The heterogeneous correlations are

positive or negative, indicating the concord or inverse relationship between the two fields. The two-tailed Student's t-test is used for statistical significance testing for the heterogeneous correlation, and the regions of significant correlations are regarded as the key areas of interaction between the two fields. For the detailed methodology of SVD calculations, please refer to Prohaska (1976). In this study, the SVD method was used to capture the coupled patterns of the spatio-temporal variations in Q1 over the TP and STT frequency over China and surrounding regions (70–135°E and 20–55°N).

## 3 The TP thermal effect on ozone STT

### 3.1 Monthly variations in STT frequency

The spatial patterns of the monthly mean of STT frequency over 1980–2014 are shown in Fig. 1. The STT frequency presents distinct monthly variations with the seasonal shift of high-frequency areas over Asia. From January to March, high STT frequencies are distributed in a zonal band south of the TP between 25°N and 35°N with the high-frequency zone extending east-west. Another zonal area of high STT frequency between 40°N and 50°N progressively expanding from the north of the TP to the regions between Northeast China and the southern part of Japan (Fig. 1a–1c), forming the northern and southern branches of the high-frequency zone around the TP.

During late spring and early summer from April to June, the southern branch of high-frequency band contracts gradually from east to west and almost vanishes in July, while the northern branch of high STT frequency over the region from Northeast China to Japan expands remarkably between 40°N to 55°N from east to west and the high STT frequency occupies the regions north of 35°N over China and the surroundings (Fig. 1d–1g), which could reflect the high period of STT from April to June over China and the surrounding Asian region.

From August to September, both the magnitudes and areas of two branches of STT frequency diminish significantly (Fig. 1h and 1i), reaching their minimum of STT seasonal cycle during the autumn months of September, October, and November (Fig. 1i–1k), which could be the period of low air mass STT throughout the year over China. Subsequently, as the STT seasonal shifts from autumn to winter, the STT recovery emerges in December over China, commencing in the region between northeast China and Japan as well as in the southern edge of the TP (Fig. 1l), leading to the resurgence of northern and southern branches of high-frequency STT zone in the subsequent year (Fig. 1a and 1b). It is worthwhile to show the season cycle of the westerly jets in association with the seasonal cycle of the STT frequency in Fig. 1. The STT frequency is notably linked to the strengths and positions of the northern and southern branch westerly jets around the TP, and the seasonal northward jump and southward retreat of the westerly jets from summer to winter significantly modulate the zonal variation of the high STT frequency in spatial distribution and seasonal patterns (Fig. 1).

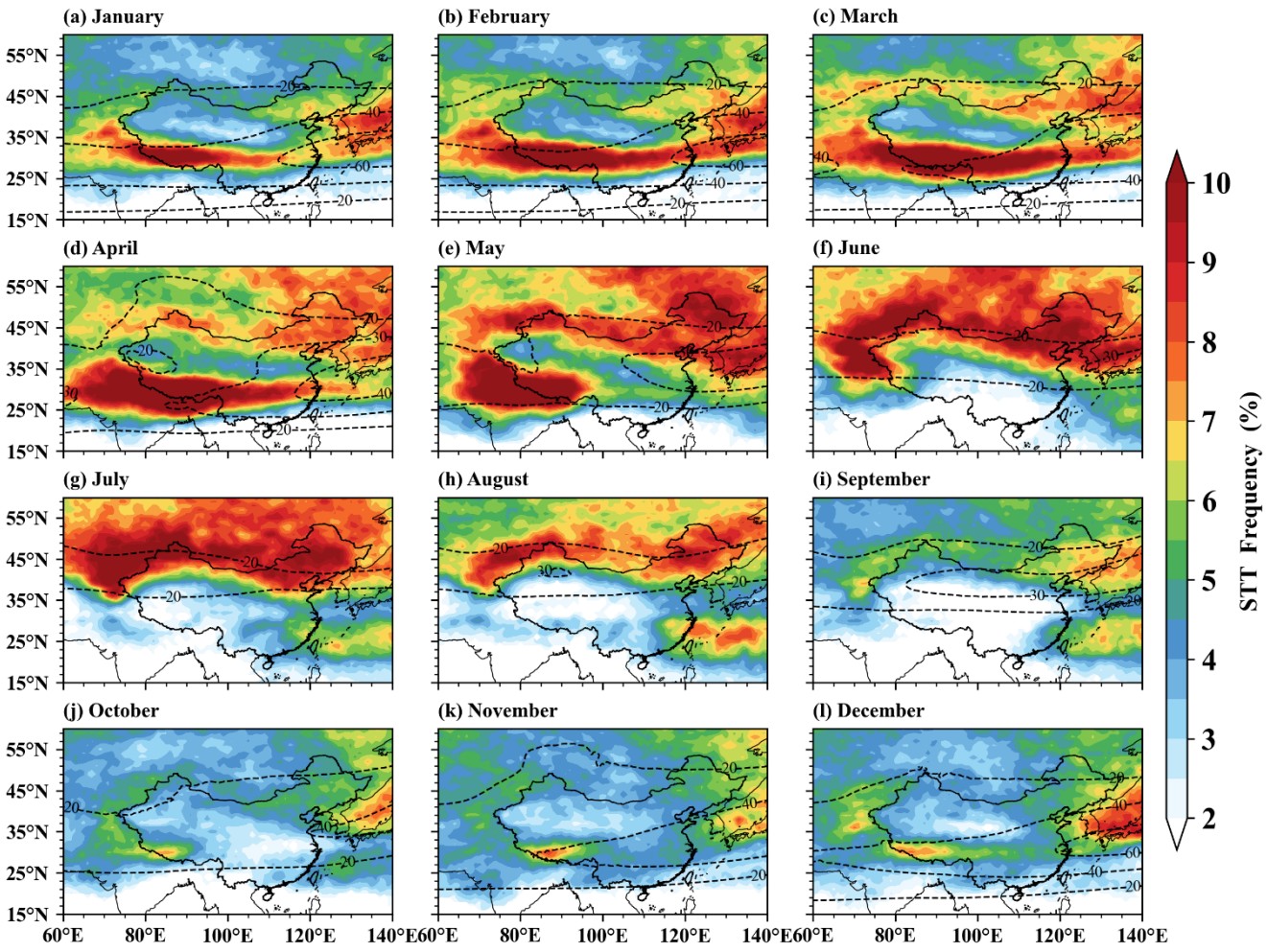

**Figure 1: Monthly variations in STT frequency (color contours) and zonal wind velocity in UTLS (contour lines) over China and the surrounding Asian region averaged during 1980–2014. The zonal wind velocity (m s$^{-1}$) is averaged over 200–300 hPa indicating the westerly jet strengths and positions.**

The STT exhibits a prominent seasonal cycle. Specifically, in late spring to early summer (April, May, and June, hereafter referred to as AMJ), the northern and southern branches of the high STT zone cover the largest area of China and surrounding areas. As illustrated in Fig. 2, The northern band, between 35 N and 55 N, stretches from western China along the northern branch of the westerly jet to southern Japan, with a large area of high STT values found between Northeast China and the Korea-Japan region, and the southern band is located between 25 N and 35 N, spanning from the western edge of the TP and the southern Asian region, southern China to the East China Sea. The two branches converge over the northwestern Pacific, consistent with the spatial distribution of the upper-tropospheric westerly jet (Cressman, 1981). The STT frequency in these two high-value bands exceeds 7%, with broad coverage significantly impacting tropospheric ozone

changes in China. The study by Yang et al. (2016) indicated that the ozone STT flux peaks in the boreal extratropical regions exist during late spring and early summer, consistent with the high-frequency STT period identified in this study.

The notable seasonal cycles of STT frequency over China and the surroundings are closely tied to the seasonal shifts of the westerly jet (Akritidis et al., 2016; Lee et al., 2024; Stohl et al., 2003), which is primarily dependent on the mechanical and thermal forcing of the TP topography with the distinct seasonal change (Wu et al., 2007, 2018). As a result, by targeting AMJ of late spring and summer, this study explores the effects of the TP thermal forcing on interannual variations in STT frequency over China.

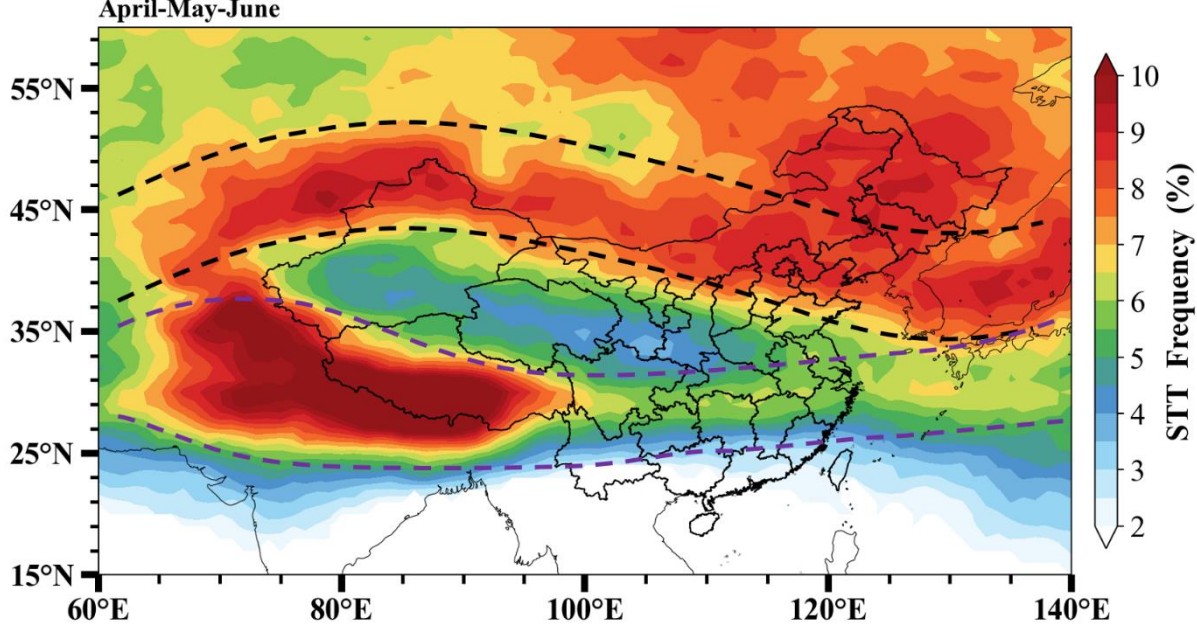

**Figure 2: The AMJ-averaged STT frequency over China and the surroundings during 1980–2014. The black and purple dashed lines denote the northern and southern branches of high STT frequency around the TP.**

**3.2 The TP thermal effect on STT frequency**

Here, we use the SVD method to explore the spatiotemporal relationship between the thermal forcing of the TP and STT frequency over China during AMJ of 1980–2014. As shown in Fig. 3 and 4, two leading modes are decomposed by SVD with variance contributions of 31.59% for SVD1 (Fig. 3a–3c) and 25.20% for SVD2 (Fig. 4a–4c), respectively. The first two pairs of leading patterns can explain 56.79% of the total variance, characterizing the majority of the relationship between the

185 TP thermal forcing and STT frequency over China.

Fig. 3a–3c present the distribution of heterogeneous correlations of TP thermal forcing and STT frequency over China, and the normalized time series of temporal coefficients of the first SVD mode. As exhibited in Fig. 3a, The heterogeneous correlations are predominantly positive over the TP, and a key area of TP thermal effect on STT change over China is identified over the central and eastern TP with significant heterogeneous correlations. While for STT frequency variations (Fig. 3b), there is an obvious zonal pattern with significantly positive correlations along the northern branch of westerly jet over northern China between 35 °N–45 °N (red dashed line in Fig. 3b), and negative correlations in the lower latitudes (20–30 °N) from the TP to southeastern China along the southern branch of westerly jet (blue dashed line in Fig. 3b) (Kuang et al., 2007). Their temporal coefficients show consistent interannual fluctuations (Fig. 3c) with a high correlation coefficient of 0.81, passing the 0.01 significance level, implying the strong effect of TP thermal forcing on interannual variations over China. The spatial distributions of heterogeneous correlations in SVD1 reveal that the interannual positive anomalies of the heat sources over the central and eastern TP are tightly associated with the positive anomalies in the STT frequency along the northern branch of westerly jet over northern China and the downturns anomalies along the southern branch over the lower-latitude regions of China, and vice versa, indicating the reverse effect of the TP thermal forcing on STT frequency variations over northern and southern China.

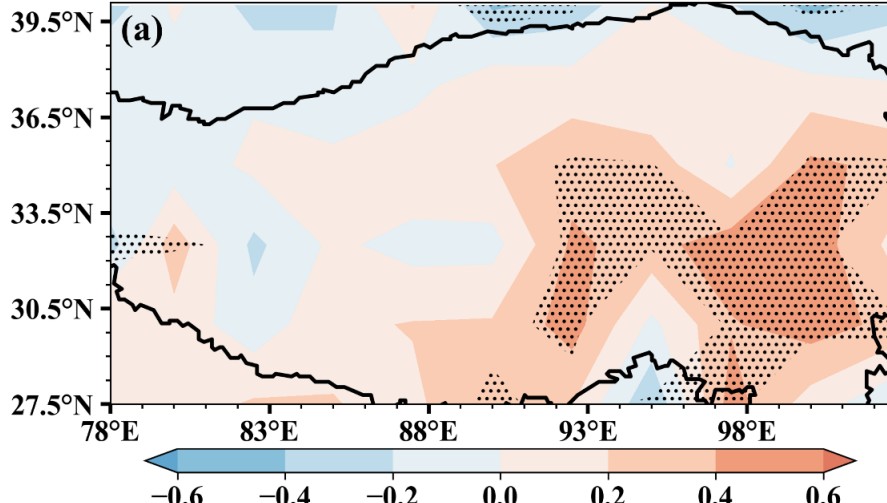

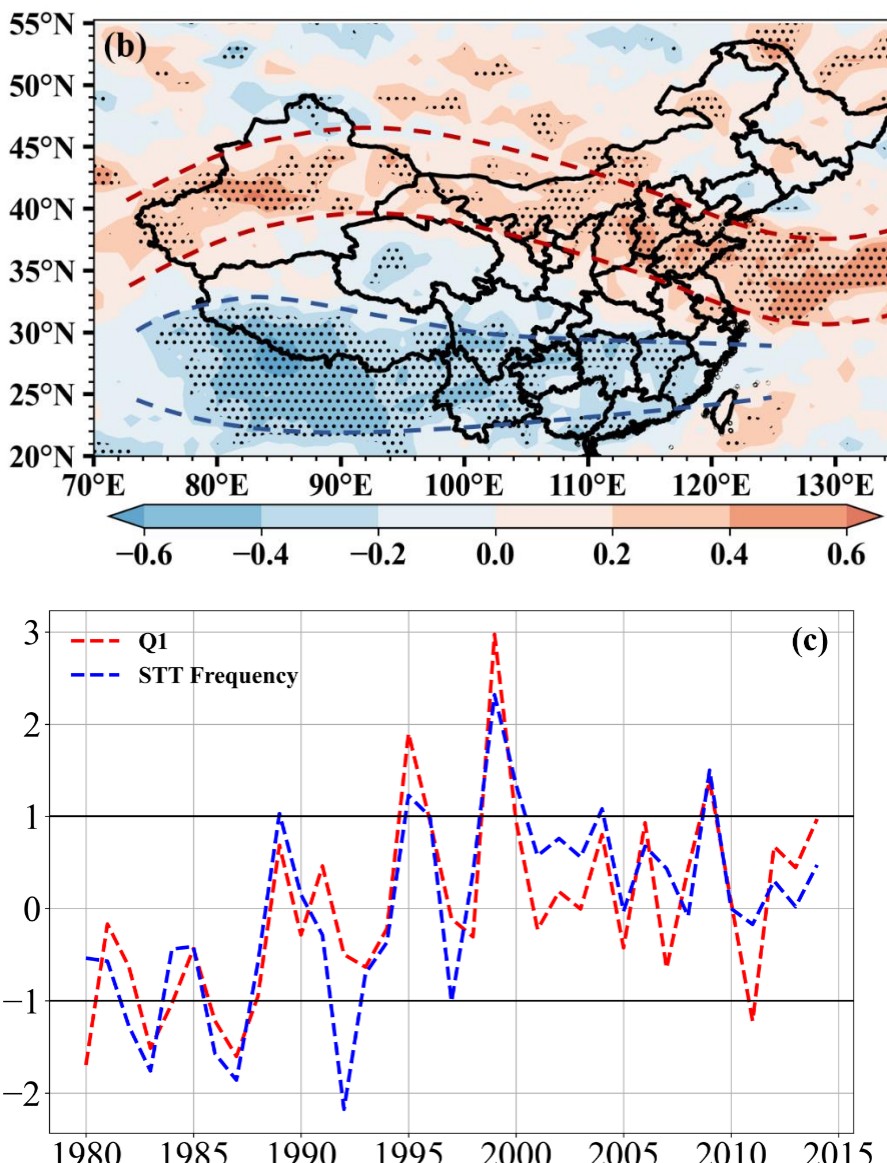

**Figure 3: Heterogeneous correlations of (a) TP thermal forcing, (b) STT frequency, where the red and blue dashed lines denote the northern and southern branches of STT frequency, respectively, and (c) their normalized temporal coefficients in the first SVD mode (SVD1). Black dots mean passing the 90 % confidence level based on the Student's t-test.**

Compared with the first mode of SVD, the second mode presents different heterogeneous correlation patterns and normalized time series of temporal coefficients (Fig. 4). The heterogeneous correlations of the TP thermal forcing are featured with the significantly positive values centered over the western TP and the negative values over small areas in the northeastern TP and southeastern edge of the TP (Fig. 4a), which are recognized as the key regions for the western

enhancing and eastern declining STT over China and the Asian region (Fig. 4b), especially the positive heterogeneous correlations of STT frequency are concentrated over North China Plain in eastern China with the significantly positive values (Fig. 4b), indicating that the North China plain is the key region of STT variations influenced by the key regions of TP thermal forcing in the second mode.

The dynamic atmospheric processes leading to stratospheric intrusions are complex. In addition to the Brewer-Dobson circulation, tropopause folds and upper tropospheric vortex processes play a significant role. A large amount of STT occurs through atmospheric dynamics in tropopause folds with mid-latitude westerly trough and the cut-off lows in the upper tropospheric vortex processes (Ancellet et al., 1994; Li et al., 2015; Meng et al., 2022; Luo et al., 2024). Ozone STT events often involve the interaction of tropopause folds, long-range transport, and turbulent dynamics, spanning spatial scales of over 1000 km for transport processes and approximately 100 km for tropopause folds. The transport of stratospheric ozone in vertical layers, with transit times ranging from longer residence times (e.g., > 10 days) from thousands of kilometers away, to shorter times (lasting less than 2 days) in the troposphere can result from slow indirect STT and rapid direct STT ozone intrusions into the lower troposphere (Zanis et al., 1999, Eisele et al., 1999), and the important distinction of ozone intrusions between direct and indirect STT is also addressed in the numerical study by Škerlak et al. (2019). The impacts of direct intrusions of stratospheric ozone on central and eastern China are more pronounced at daily timescales and confined to limited areas, whereas indirect intrusions of stratospheric ozone originate from remote regions thousands of kilometers away, evenly distributed latitudinally along the westerlies between 40 ° and 70 °N (Meng et al., 2024). Considering the difference in the spatial extent of key regions with positive STT anomalies over China and surroundings between the first and second SVD modes (Fig. 3b and 4b), the key region of TP thermal forcing in the second mode could induce the more direct stratospheric intrusion with the direct STT from North China Plain to the troposphere over central and central China, while the key region of TP thermal forcing in the first mode could build the indirect pathways of stratospheric ozone intrusions to the troposphere over central and eastern China indirectly from thousands of kilometers away from the key STT along the extratropical westerly jet. The regional anomalies in TP thermal forcing could alter the direct and indirect pathways of stratospheric ozone intrusions to the troposphere over China.

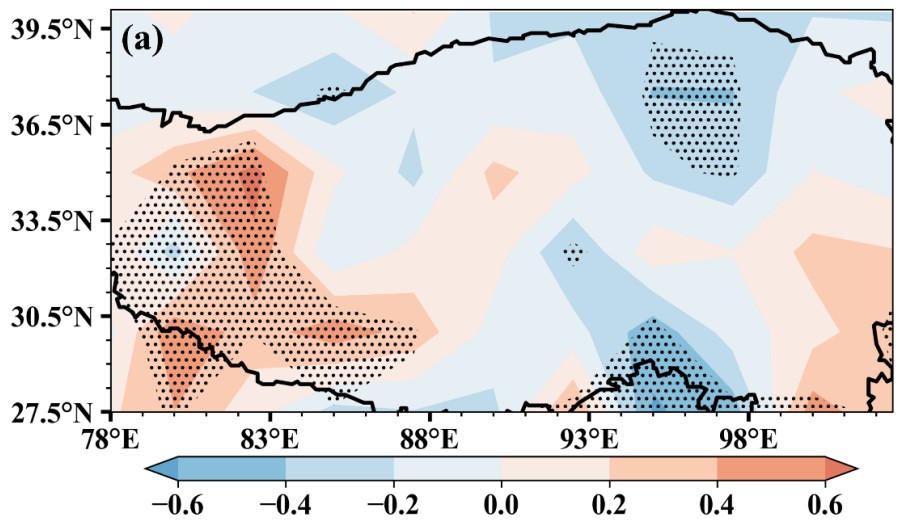

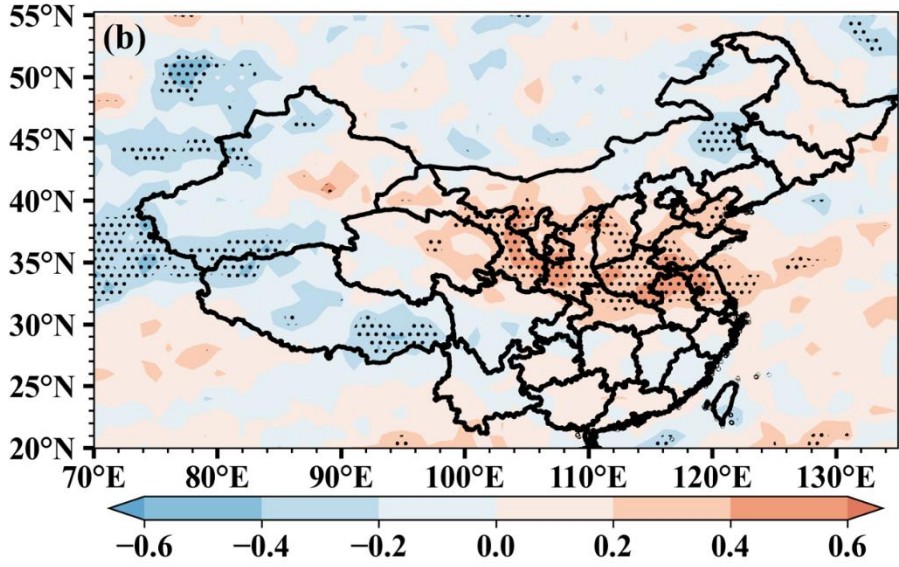

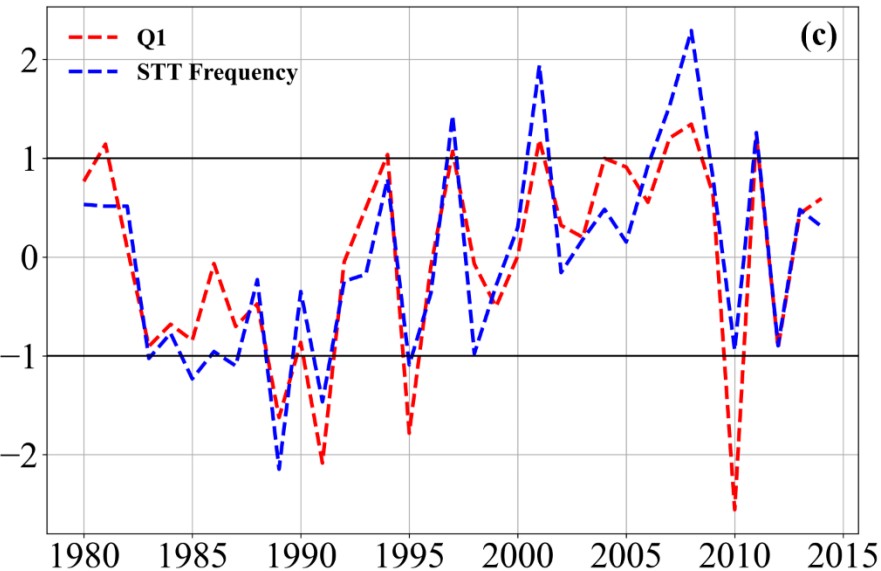

Figure 4: Same as Fig. 3, but for the second SVD mode (SVD2).

Although TP covers a smaller region compared to the broader Asian monsoon region, the thermal forcing over the TP, as "the world roof", is the decisive factor in the formation of the Asian monsoon anticyclone and plays a significant role in building the Asian monsoons (Wu et al., 2015; Duan et al, 2024; Xu et al., 2010). Figs. 3a and 4a encompass the region from 78 °–103 °E and 28 °–38 °N, covering most areas with elevations exceeding 3000 m, which are identified as the extent of the TP (Xu et al., 2016). Moreover, the areas with heterogeneous correlations passing the significance test are primarily concentrated over the TP region including the TP platform within the 3000 m elevation border (black line) and some regions of the TP slopes near the TP platform. Therefore, our study focuses on this region to investigate the relationship between TP thermal forcing and STT variations. Previous studies have indicated that the thermal effects of the TP influence the weather and climate patterns in China by modulating the strength and position of the westerly jet (Wu et al., 2007; Liu et al., 2020; Tan et al., 2023). The first mode of SVD accurately captures the changes in STT occurrences along the northern and southern branches of westerly jet and the connection with the TP heat sources. Considering that SVD1 is the predominant decomposition mode, accounting for 31.59% of the total variance, representing the main change pattern, our analysis hereby focuses on the ozone STT flux changes as depicted by SVD1 and explores the meteorological mechanism of the cause-effect relationship in the following sections.

### 3.3 Anomalies of ozone STT flux induced by the TP thermal forcing

The AMJ-averaged ozone STT flux during 1980–2014 has a spatial distribution similar to the STT frequency over China (Fig. 2). In general, the high values of ozone STT flux are concentrated along the upper-tropospheric westerly jet during AMJ over China including the NCP, northeast China, and northwestern parts of China as well as in the southern and western TP edges, while the lower ozone STT flux values are located in southern China, exhibiting an overall spatial pattern of two branches of high ozone STT flux along the southern and northern branches of westerly jet around the TP (Fig. 5a).

A composite analysis was utilized to characterize the spatial patterns of ozone STT flux change with the TP thermal anomalies. Based on the normalized temporal coefficients of TP thermal forcing in SVD1 (Fig. 3c), the years with Q1 greater and less than one standard deviation during AMJ were selected as the positive and negative Q1 anomaly years, respectively. Thus, four positive Q1 years (1995, 1999, 2009, and 2014) and six negative Q1 years (1980, 1983, 1984, 1986, 1987, and 2011) are selected for the following composite analysis on anomalies of ozone STT flux induced by the TP thermal forcing.

The reverse patterns of spatial distributions in the composite anomalies of ozone STT flux over China are presented in positive and negative Q1 years (Fig. 5b and 5c). The positive anomalies of TP thermal forcing induce the positive anomalies of ozone STT flux along the northern branch westerly jet over northern China and negative ones in central and southern China along the southern branch (Fig. 5b), whereas the negative anomalies of TP thermal forcing trigger the opposite anomalies of ozone STT flux over China comparing to the positive anomalies of TP thermal forcing (Fig. 5c).

The differences in the ozone STT flux anomalies during AMJ between positive and negative Q1 years exhibit a prominent spatial dipole pattern, characterized by positive anomalies in the north and negative anomalies in the south over China, with large areas passing the significance test (Fig. 5d), confirming spatial patterns of TP thermal forcing influencing the STT of ozone over China revealed by the SVD analysis of STT frequency (Fig. 3b). It is noteworthy that the large positive anomalies exist in northern China with climatologically high ozone STT flux, while negative anomalies are centered in southern China with relative low ozone STT flux (Fig. 5a and 5d), indicating that the strong TP thermal forcing exerts the reverse impacts on interannual variations of ozone STT along the southern and northern branches of westerly jet with exacerbating the north-south gradient of ozone STT flux.

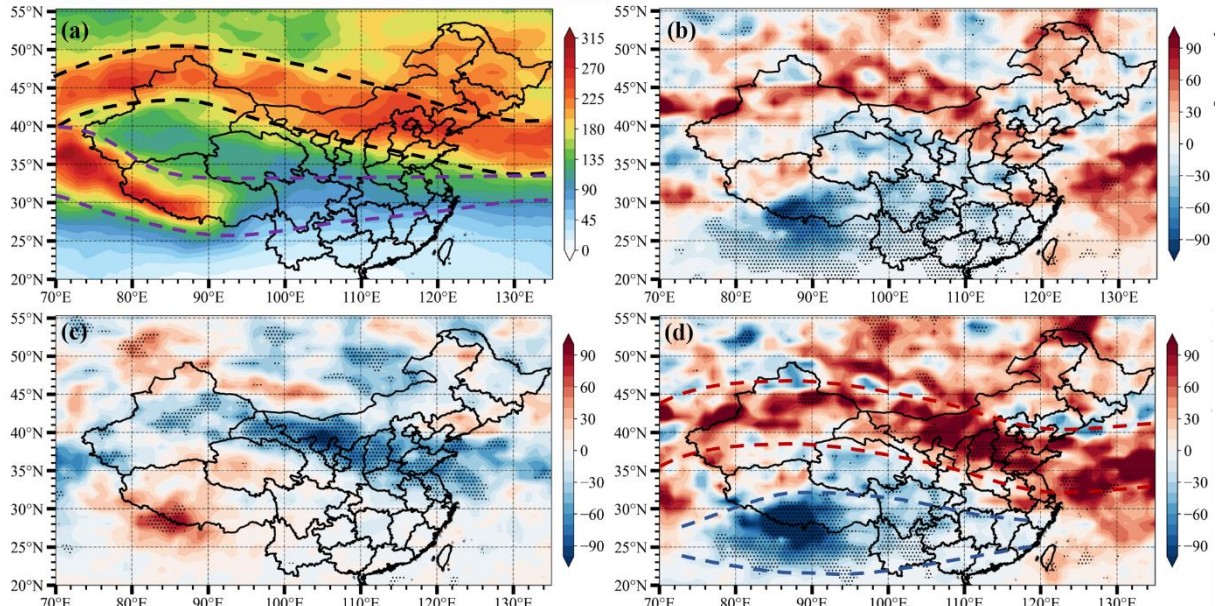

**Figure 5: (a) The AMJ-averaged ozone STT flux (kg km$^{-2}$ month$^{-1}$) during 1980-2014. Composite anomalies of ozone STT flux (kg km$^{-2}$ month$^{-1}$) during (b) positive years, (c) negative years, and (d) their differences. Dashed lines denote the northern and southern branches of ozone STT flux, respectively. Black dots indicate the composite anomalies passing the 90 % confidence level based on the Student's t-test.**

To further quantify the impact of TP thermal forcing on ozone STT flux, we calculated the regional averages of ozone STT flux changes over the northern branch (78–120°E, 35–45°N) with significant positive anomalies and the southern branch (78–120°E, 21–31°N) with prominent negative anomalies (Fig. 5d). The AMJ regional average of ozone STT flux in the northern branch reach up to 180.34 kg km$^{-2}$ month$^{-1}$, which is over twice the regional value of 73.71 kg km$^{-2}$ month$^{-1}$ in the southern branch (Table 1). With the most positive anomalies of TP thermal forcing, the composite anomalies in ozone STT flux are estimated with 14.73 kg km$^{-2}$ month$^{-1}$ in the increase rate at 8.17% in the northern branch, and 23.23 kg km$^{-2}$ month$^{-1}$ in the reduction rate at 31.52% in the southern branch relative to the climatological AMJ averages (Table 1). The positive anomalies of the TP thermal forcing could enhance ozone STT over northern China and decline ozone STT over southern China. In contrast, the composite anomalies of ozone STT flux present the opposite patterns over China during AMJ with the most negative anomalies of TP thermal forcing, which is estimated that the regional decrease at 24.64 kg km$^{-2}$ month$^{-1}$ in the change rate of -13.66% in the northern branch, and the increase of 7.54 kg km$^{-2}$ month$^{-1}$ in the change rate of 10.23% in the southern branch (Table 1). The negative anomalies of the TP thermal forcing could weaken ozone STT flux over northern China and aggravate ozone STT over southern China.

**Table 1** The averages (kg km$^{-2}$ month$^{-1}$), the changes (kg km$^{-2}$ month$^{-1}$), and the relative contributions (changes/averages in %) of ozone STT flux over the northern (78–120 °E, 35–45 °N) and southern (78–120 °E, 21–31 °N) branches induced by the most positive and negative anomalies of TP thermal forcing during AMJ of 1980–2014.

| | Northern branch | Southern branch |
|---|---|---|
| AMJ climate means | 180.34 | 73.71 |
| Positive AMJ anomalies | 14.73 (8.17%) | -23.23 (-31.52%) |
| Negative AMJ anomalies | -24.64 (-13.66%) | 7.54 (10.23%) |

### 3.4 Meteorological mechanism of TP thermal effect on STT

The thermal effect of TP can generate large-scale vertical convection with a shallow cyclonic circulation in the lower troposphere and deep anticyclonic circulation in the UTLS (Duan and Wu, 2005; Wu, 2000), playing a decisive role in the formation and maintenance of the South Asian High and changes in upper-level wind fields (Ge et al., 2018; Liu et al., 2001; Ren et al., 2019). Since the STT frequency and ozone flux over China are closely linked to the westerly jet (Hsu et al., 2005, Lee et al., 2024), the changes in the upper-level westerly jet could be an important meteorological mechanism of the thermal effect over the TP on ozone STT flux over China.

As is shown in Fig. 6, the anomalous strong thermal forcing of the TP triggered the UTLS circulation pattern (200 hPa) with the anticyclonic anomalies over the TP and the cyclonic anomalies in northeastern China, generating anomalous westerlies to the north of the anticyclone and anomalous easterlies to the south of the TP (Fig. 6a), increasing and weakening the intensity and meridional structure of the northern and southern branches of westerly jet, respectively. Subsequently, such a pattern of anomalous structures of the upper-level westerlies enhances (weakens) the tropopause folds (Fig. 6b) along the northern (southern) branch of the westerly jet for more (less) STT frequency and ozone flux over China and surrounding regions.

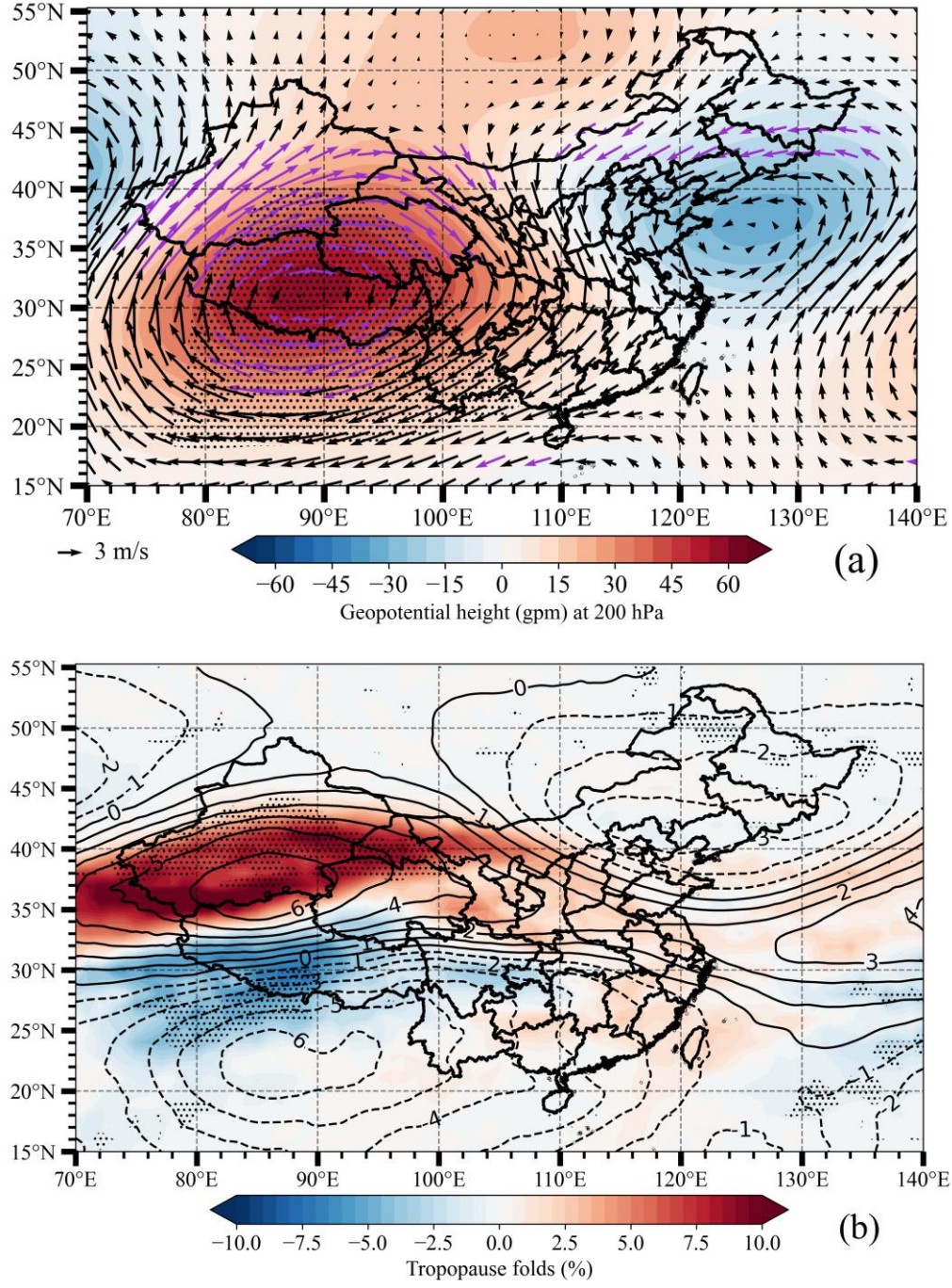

**Figure 6: Composite differences in (a) geopotential height (shaded, gpm) and wind vectors at 200 hPa, (b) tropopause fold frequency (shaded, %) and zonal wind speed at 200 hPa (contour, m/s) with solid (dashed) contours representing an enhancement (attenuation) in westerly jet. Black dots and purple vectors indicate the composite anomalies passing the 90 % confidence level based on the Student's t-test.**

Potential vorticity (PV) and relative humidity (RH) are essential meteorological indicators for STT in tropopause folds. A

typical tropopause fold is always characterized by elevated PV values, and decreased RH values near or below the

tropopause (James et al., 2003; Kim and Lee, 2010). Since the interannual variations in AMJ-averaged TP thermal forcing

can change the intensity and meridional structures of upper-tropospheric westerly jet for the tropopause folds, the

meteorological mechanism of the TP thermal forcing affecting STT could be explored with the PV and RH changes in the

UTLS induced by the atmospheric heating anomalies over the TP.

The composite PV and RH differences averaged in the UTLS between the positive and negative anomaly years of the TP

thermal forcing exhibited similar spatial patterns to the heterogeneous correlations of STT frequency in SVD1 (Fig. 3b) and

the anomalies of ozone STT ozone flux over China (Fig. 5d), which are induced by the changes of TP thermal forcing. There

are significant positive (negative) PV (Fig. 7a) and negative (positive) RH anomalies (Fig. 7b) in the UTLS over China along

the northern (southern) branch of westerly jet, revealing the meteorological mechanisms of STT changes with the tropopause

folds affected by the TP thermal anomalies.

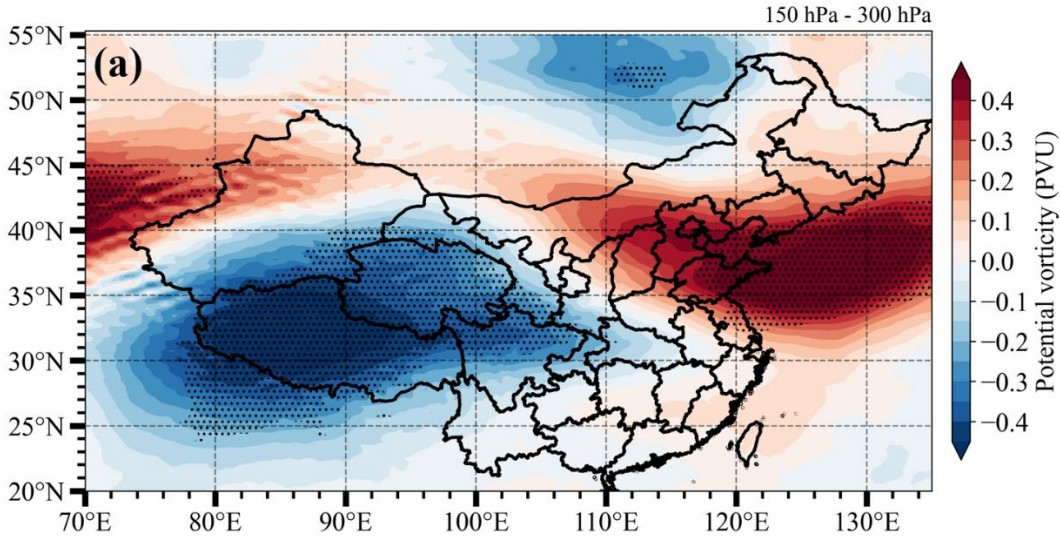

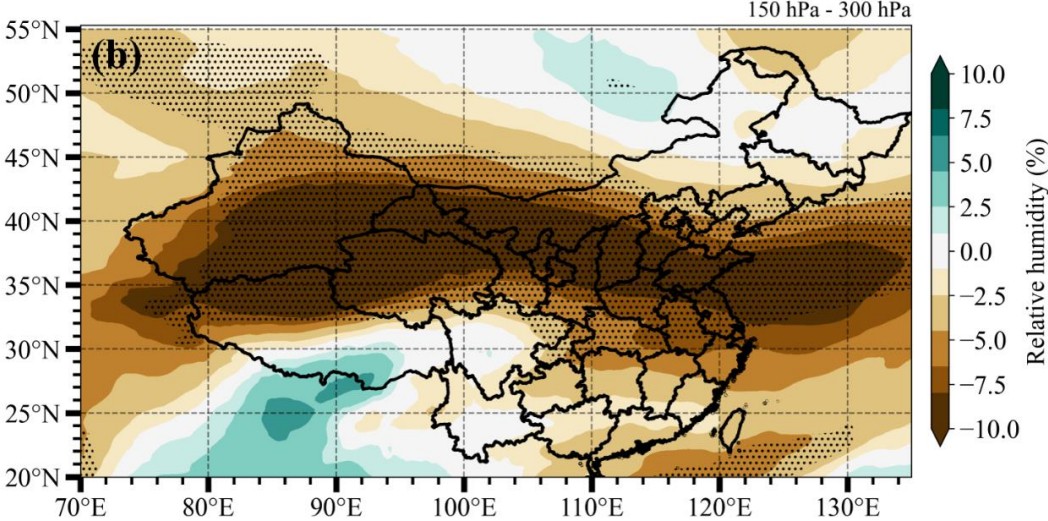

**Figure 7:** Composite differences in (a) potential vorticity (PVU) and (b) relative humidity (%) averaged in the UTLS from 150 hPa to 300 hPa between positive and negative Q1 anomaly years. Black dots indicate the composite anomalies passing the 90 % confidence level based on the Student's t-test.

## 4. Conclusions

The TP topographic forcing plays an important role in global and regional climate as well as environment changes. However, the TP thermal effect on the interannual changes in ozone STT over China has not been extensively studied. Therefore, in this study, based on the STT frequency and ozone flux dataset and ERA5 data meteorological reanalysis for 1980–2014, the interannual variations in STT of ozone over China induced by the TP thermal forcing were investigated. The main results are concluded as follows.

The STT frequency in the Asian regions exhibits a distinct seasonal cycle with monthly changes. High STT frequency is concentrated in late spring and early summer of April, May, and June, covering areas along the northern and southern branches of westerly jet around the TP over the Asian region. Through SVD diagnostic analysis, we identified a synergistic relationship between interannual variations of TP thermal forcing and STT frequency. The intensifying thermal forcing in the central and eastern TP is closely linked to increasing and decreasing STT frequency respectively along the northern and southern branches of westerly jet, while the enhancing thermal forcing over the western TP is related to the western enhancement and eastern decline STT over China and surrounding regions. The thermal forcing changes over the TP regions could regulate the interannual STT anomalies over China by altering the direct and indirect pathways of ozone STT. This study reveals a unique driver of the TP thermal forcing on the interannual variations of ozone STT with diverse structures, providing an insight into the TP effect on atmospheric environment and climate changes.

Stronger TP thermal forcing leads to an increase in ozone STT flux by 14.73 kg km⁻² month⁻¹ (8.17%) along the northern branch and a decrease of 23.23 kg km⁻² month⁻¹ (31.52%) along the southern branch of westerly jet over China, while weak TP thermal forcing induces the ozone STT flux decrease of 24.64 kg km⁻² month⁻¹ (13.66%) in the northern branch and the increase of 7.54 kg km⁻² month⁻¹ (10.23%) in the southern branch, indicating the contrasting effects of TP thermal anomalies on the regional ozone STT changes in northern and southern China.

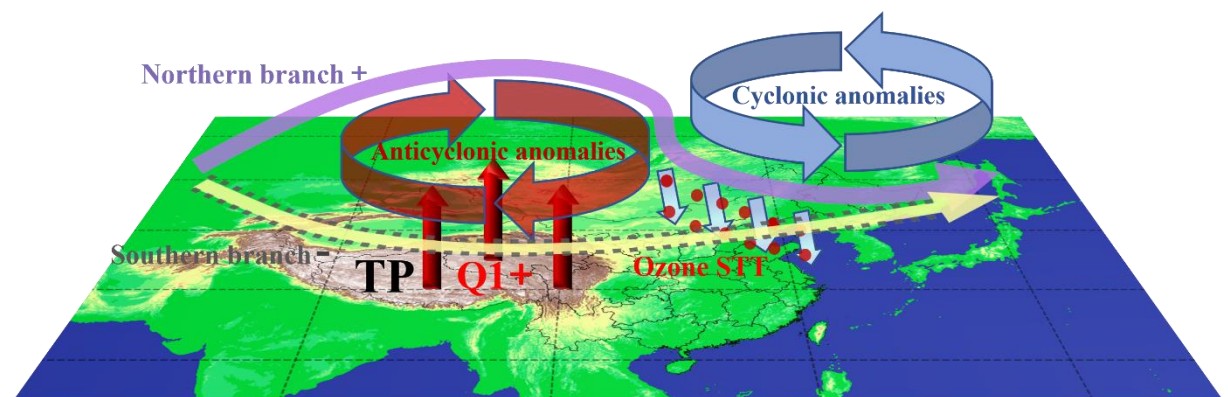

**Figure 8: Diagram of meteorological mechanism on ozone STT change along the southern and northern branches of westerly jet over China and the surrounding Asian region with anticyclonic and cyclonic anomalies in the UTLS induced by the Tibetan Plateau (TP) thermal forcing (Q1), where '+' and '-' indicating enhancement and attenuation, respectively.**

The meteorological mechanism of TP thermal effects on ozone STT along the southern and northern branches of westerly jet around the TP was proposed in Fig. 8. The intensifying thermal forcing over the TP could induce an anomalous anticyclonic circulation in the UTLS over the TP and an anomalous cyclonic circulation in northeastern China. The anomalous circulations upraise the ridge of the northern branch deepening the downwind East Asia trough along the westerly jet, and fill the southern branch trough of westerly jet, which could alter the intensity and structure of the northern and southern branches of the westerly jet, respectively, enhancing and weakening the tropopause folds along the northern and southern branches of westerly jet for more and less STT of ozone over China and surrounding regions.

The distinct structures of interannual STT anomalies with the key areas of TP thermal forcing are identified based on the 35-year climatological statistics in this study with the implication of tropospheric ozone change with the TP effects. In future studies, the effects of multiple climatic factors with multiscale variations could be considered to comprehend our understanding. Moreover, the fine observational network over the TP complex terrain could improve the assessment of the TP thermal forcing and regional climate change. Further investigation on climate modulation of TP on ozone STT requires updated climate models with comprehensive processes of atmospheric physics and chemistry.

*Data availability.* Data used in this paper can be provided upon request from Qingjian Yang (yangqj@nuist.edu.cn) or

390 Tianliang Zhao (tlzhao@nuist.edu.cn).

*Author contributions.* QY: Data curation, Methodology, Investigation, Writing - original draft. TZ: Conceptualization, Methodology, Writing - review & editing. YB and KM: Conceptualization, Methodology. YL, ZT, XS, WF, KY, and JH: Investigation, Conceptualization. All authors commented on the paper.

*Competing interests.* The authors declare that they have no conflict of interest.

*Acknowledgement.* This research was supported by the National Key Research and Development Program of China (2022YFC3701204), the National Natural Science Foundation of China (42275196, 42475195), and the Postgraduate

Research and Practice Innovation Program of Jiangsu Province (KYCX23_1317).

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
