# Peer review of "Distinct structures of interannual variations in stratosphere-totroposphere ozone transport induced by the Tibetan Plateau thermal forcing"

_EGUsphere, 2025_

## Referee Comment (RC1)

**Review for**

*Distinct structures of interannual variations in stratosphere-to-troposphere ozone transport induced by the Tibetan Plateau thermal forcing*

by Yang et al.

**Summary:**

In their study, Yang et al. consider stratosphere-to-troposphere (STT) ozone transport over the Tibetan Plateau (TP). They distinguish between direct and indirect STT processes, and link the two to the thermal forcing of the TP. They make extensive and excellent use of a STT climatology and meteorological features (e.g., tropopause folds) provided by the atmospheric dynamics group of ETH, and combine these with their own meteorological analysis, mostly considering jet streams and TP thermal forcing.

The study is certainly of high interest, as the TP is a hotspot of cross-tropopause mass and ozone transport.  The manuscript is (mostly) very well written and follows a clear storyline. First, the monthly variation in STT frequency is studied. Then, the effect of TP thermal forcing on STT frequencies is assessed, where the corresponding jet stream variability as a key driver for STT due to the thermal forcing is determined. To this aim, a singular-value decomposition (SVD) is performed to get the 'correlation' between two meteorological fields. In a next step, the ozone STT fluxes are studied, in relation to the TP thermal forcing. Finally, the meteorological drivers for STT fluxes and how they relate to the TP thermal forcing is determined. To this aim, the large-scale circulation over the TP is studied, and it is also determined how the frequency of tropopause folds varies in response to a varying TP forcing. The  manuscript concludes with a nice schematic figure that summarizes the key findings.

It was fun reading the manuscript, in particularly to see how the text is very well structured and leads the reader step-by-step to the answers of the key questions raised in the introduction. I can definitely recommend publication of the paper with minor revisions. There are only few specific comments that the authors could address to improve readability.

**Specific comments:**

- The term 'Tibetan Plateau thermal forcing' is used in the abstract and also in the introduction (L62), but there it is not clearly defined. Possibly, the authors could add a few sentences that introduces the meaning of 'thermal forcing' more clearly. This might be important, as the TP thermal forcing is a key aspect of the study.

- L28-29 [This study … environment]: I think I understand what the authors mean, but it could be phrased more clearly.

- L35: 'Its abundance being...' -> ?its abundance in the troposphere being'

- Figure 1: The STT frequency seems to be strongly linked to the strength and position of the jet streams around and over the TP. I wonder whether it would be worthwhile to show the season cycle of the jet streams in addition to the seasonal cycle of the STT frequency. For example, in Figure 1 one could get (most likely) remove the detailed country(canton boundaries, and instead overlay some wind speed contours. In this way, the close link between jets and STT frequency could be highlighted.

- L146: 'The STT exhibits a prominent seasonal cycle with monthly changes'. -> What does 'with monthly changes' exactly mean? I assume that this reflects a slight 'language uncertainty' (not being native English speakers?), although the text is mostly well written and clear. Still, at some places a further language editing could help to make the text more concise and clearer.

- L160: Similarly, 'with the underlying meteorological mechanisms' is not clear. Which meteorological mechanisms?

- In Section 3.2 the authors might make clearer that the SVD-based analysis does not yet imply a cause-effect relationship. It is, as far as I understand, only a measure of 'correlation' between two fields. The text, however, already points to such a cause-effect relationship. This will only later in the text be justified when the jet variability is 'explained' by the thermal forcing.

- L203: 'and upper tropospheric vortex processes play a significant role' -> It is not completely clear to me what 'vortex processes' the authors are referring to. Some further explanations (more explicit naming of processes) would be helpful.

- In Section 3.2, a distinction is made between direct STT (with intense vertical transport) and indirect STT (with longer-range transport). I wonder whether this relates also to the STT process that is discussed in detail in the following study:

> *Škerlak, B., Pfahl, S., Sprenger, M., and Wernli, H.: A numerical process study on the rapid transport of stratospheric air down to the surface over western North America and the Tibetan Plateau, Atmos. Chem. Phys., 19, 6535–6549, https://doi.org/10.5194/acp-19-6535-2019, 2019.*

At least, it seems to me that the important distinction between direct and indirect STT is also addressed in this numerical study. If so, the authors might want to refer to this study; if not, of course, there is no need to include it. I think it would be worthwhile to include some additional studies regarding direct/(indirect STT, as the discussion in L202-215 is certainly very plausible, but it is also somewhat more speculative than other parts of the paper.

- The analysis in Section 3.4 about meteorological mechanism is very nice, and clearly addresses some of my earlier concerns about cause-and-effect vs. simple correlation. Possibly, the authors can earlier in the manuscript already point to this section where good reasons for a cause-effect relationship is made.

---

## Author Comment (AC1)

Dear Editors and Referees:

Thank you very much for your careful review and constructive comments on our manuscript acp-2025-737. We have accordingly revised the manuscript carefully. The revised portions are highlighted in the revised manuscript. In the followings, we quoted each review question in the square brackets and added our response after each question.

**Responses to Reviewer #1**

*[General comments: In their study, Yang et al. consider stratosphere-to-troposphere (STT) ozone transport over the Tibetan Plateau (TP). They distinguish between direct and indirect STT processes, and link the two to the thermal forcing of the TP. They make extensive and excellent use of a STT climatology and meteorological features (e.g., tropopause folds) provided by the atmospheric dynamics group of ETH, and combine these with their own meteorological analysis, mostly considering jet streams and TP thermal forcing. The study is certainly of high interest, as the TP is a hotspot of cross-tropopause mass and ozone transport. The manuscript is (mostly) very well written and follows a clear storyline. First, the monthly variation in STT frequency is studied. Then, the effect of TP thermal forcing on STT frequencies is assessed, where the corresponding jet stream variability as a key driver for STT due to the thermal forcing is determined. To this aim, a singular-value decomposition (SVD) is performed to get the 'correlation' between two meteorological fields. In a next step, the ozone STT fluxes are studied, in relation to the TP thermal forcing. Finally, the meteorological drivers for STT fluxes and how they relate to the TP thermal forcing is determined. To this aim, the large-scale circulation over the TP is studied, and it is also determined how the frequency of tropopause folds varies in response to a varying TP forcing. The manuscript concludes with a nice schematic figure that summarizes the key findings. It was fun reading the manuscript, in particularly to see how the text is very well structured and leads the reader step-by-step to the answers of the key questions raised in the introduction. I can definitely recommend publication of the paper with minor*

*revisions. There are only few specific comments that the authors could address to improve readability.]*

**Response:** Many thanks for the encouraging comments and helpful suggestions on our manuscript. Following the reviewer's suggestions and comments, we have accordingly made careful revisions. Please find our point-to-point responses as follows:

*Specific comments:*

1. *[The term 'Tibetan Plateau thermal forcing' is used in the abstract and also in the introduction (L62), but there it is not clearly defined. Possibly, the authors could add a few sentences that introduces the meaning of 'thermal forcing' more clearly. This might be important, as the TP thermal forcing is a key aspect of the study]*

**Response 1:** Thanks for your suggestion. We have added the description of Tibetan Plateau thermal forcing in the revised Sec. 1 Introduction (lines 63–67) as follows:

"The TP serves as a large heat source in boreal summer because high surface sensible heating could cause cyclonic circulation anomalies in the lower troposphere and could trigger vertical motion with notable latent heat release from cloud-precipitation in the middle-upper troposphere over the TP (Luo and Yanai, 1984; Duan and Wu, 2005; Wu et al., 2007). The TP thermal forcing refers to the heating intensity in the atmosphere over the TP, resulting from the atmospheric diabatic process of surface sensible heat, latent heat, and radiation heat."

**References:**

Luo, H. and Yanai, M.: The large-scale circulation and heat sources over the Tibetan Plateau and surrounding areas during the early summer of 1979. Part II: Heat and moisture budgets, Monthly Weather Review, 112, 966–989, https://doi.org/10.1175/1520-0493(1984)112<0966:TLSCAH>2.0.CO;2, 1984.

Duan, A. M. and Wu, G. X.: Role of the Tibetan Plateau thermal forcing in the summer climate patterns over subtropical Asia, Climate Dynamics, 24, 793–807, https://doi.org/10.1007/s00382-004-0488-8, 2005.

Wu, G., Liu, Y., Zhang, Q., Duan, A., Wang, T., Wan, R., Liu, X., Li, W., Wang, Z., and Liang, X.: The influence of mechanical and thermal forcing by the Tibetan Plateau on Asian climate, Journal of

2.  *[L28-29 [This study … environment]: I think I understand what the authors mean, but it could be phrased more clearly.]*

**Response 2:** We have revised the sentence as below:

"This study reveals the distinct structures of interannual variations in ozone STT induced by the TP thermal forcing, providing a new prospect for the TP effect on atmospheric environment."

3.  *[L35: 'Its abundance being…' -> ?its abundance in the troposphere being]*

**Response 3:** We have corrected "its abundance being" to "its abundance in the troposphere being".

4.  *[Figure 1: The STT frequency seems to be strongly linked to the strength and position of the jet streams around and over the TP. I wonder whether it would be worthwhile to show the season cycle of the jet streams in addition to the seasonal cycle of the STT frequency. For example, in Figure 1 one could get (most likely) remove the detailed country(canton boundaries, and instead overlay some wind speed contours. In this way, the close link between jets and STT frequency could be highlighted.]*

**Response 4:** Thanks for your suggestion. We have accordingly redrawn Fig. 1 by removing the detailed country/canton boundaries, and overlaying zonal wind speed contours to highlight the relationship between westerly jets and STT frequency and added the discussion in the revised Sect. 3.1 (lines 149–153) as follows:

"It is worthwhile to show the season cycle of the westerly jets in association with the seasonal cycle of the STT frequency in Fig. 1. The STT frequency is notably linked to the strengths and positions of the northern and southern branch westerly jets around the TP, and the seasonal northward jump and southward retreat of the westerly jets from summer to winter significantly modulate the zonal variation of the high STT frequency

in spatial distribution and seasonal patterns (Fig. 1)."

[Figure]

**Figure 1** Monthly variations in STT frequency (color contours) and zonal wind velocity in UTLS (contour lines) over China and the surrounding Asian region averaged during 1980–2014. The zonal wind velocity is averaged over 200–300 hPa indicating the westerly jet strengths and positions.

5. *[L146: 'The STT exhibits a prominent seasonal cycle with monthly changes'. -> What does 'with monthly changes' exactly mean? I assume that this reflects a slight 'language uncertainty' (not being native English speakers?), although the text is mostly well written and clear. Still, at some places a further language editing could help to make the text more concise and clearer.*

*L160: Similarly, 'with the underlying meteorological mechanisms' is not clear. Which meteorological mechanisms?]*

**Response 5:** Thanks for your careful review. In the revised manuscript, we have rephrased "The STT exhibits a prominent seasonal cycle with monthly changes" to "The STT exhibits a prominent seasonal cycle" (line 160), and corrected "this study explores the effects of the TP thermal forcing on interannual variations in STT frequency over China with the underlying meteorological mechanisms" to "this study explores the effects of the TP thermal forcing on interannual variations in STT frequency over China" (lines 173–174).

In addition, we have made the further language editing completely in the revised manuscript to make the text more concise and clearer.

6. *[In Section 3.2 the authors might make clearer that the SVD-based analysis does not yet imply a cause-effect relationship. It is, as far as I understand, only a measure of 'correlation' between two fields. The text, however, already points to such a cause-effect relationship. This will only later in the text be justified when the jet variability is 'explained' by the thermal forcing]*

**Response 6:** We agree with the reviewer's comment. The SVD method is a mathematical technique to identify the spatiotemporal correlation between two variable fields. After establishing the correlation between the two variables, the SVD-based analysis does not yet imply a cause-effect relationship, only a measure of 'correlation' between the two fields. In our study, the SVD-based analysis first identifies the spatiotemporal relationship between the thermal forcing of the TP and STT frequency over China using SVD decomposition, and then we examine the meteorological mechanism with composite analysis in Sec. 3.4 to further investigate the cause-effect relationship between TP thermal forcing and STT variations when the jet variability is 'explained' by the thermal forcing.

7. *['and upper tropospheric vortex processes play a significant role' -> It is not completely clear to me what 'vortex processes' the authors are referring to. Some further explanations (more explicit naming of processes) would be helpful]*

**Response 7:** Thanks for your suggestion. The troughs and vortices in the mid-latitude

westerlies could lead to the tropopause folds, triggering the intrusions of stratospheric ozone-rich air to the troposphere. Induced by the cut-off low, a vortex in the mid-latitude westerly jet over the Siberian region, a slanted channel of ozone downward intrusion was set up from the UTLS to the lower troposphere over the North China Plain (Meng et al., 2022); The deep stratospheric ozone intrusions over North China was driven by the the upper tropospheric vortex development along the westerly trough (Luo et al., 2024). We have clarified the 'vortex processes' in the revised manuscript (lines 217–219):

" A large amount of STT occurs through atmospheric dynamics in tropopause folds with mid-latitude westerly trough and the cut-off lows in the upper tropospheric vortex processes (Ancellet et al., 1994; Li et al., 2015; Meng et al., 2022; Luo et al., 2024)."

**References:**

Ancellet, G., Beekmann, M., and Papayannis, A.: Impact of a cutoff low development on downward transport of ozone in the troposphere, Journal of Geophysical Research: Atmospheres, 99, 3451–3468, https://doi.org/10.1029/93JD02551, 1994.

Li, D., Bian, J., and Fan, Q.: A deep stratospheric intrusion associated with an intense cut-off low event over East Asia, Science China Earth Sciences, 58, 116–128, 2015.

Meng, K., Zhao, T., Xu, X., Zhang, Z., Bai, Y., Hu, Y., Zhao, Y., Zhang, X., and Xin, Y.: Influence of stratosphere-to-troposphere transport on summertime surface O3 changes in North China Plain in 2019, Atmospheric Research, 276, 106271, https://doi.org/10.1016/j.atmosres.2022.106271, 2022.

Luo, Y., Zhao, T., Meng, K., Hu, J., Yang, Q., Bai, Y., Yang, K., Fu, W., Tan, C., Zhang, Y., Zhang, Y., and Li, Z.: A mechanism of stratospheric O$_3$ intrusion into the atmospheric environment: a case study of the North China Plain, Atmos. Chem. Phys., 24, 7013–7026, https://doi.org/10.5194/acp-24-7013-2024, 2024.

8. *[In Section 3.2, a distinction is made between direct STT (with intense vertical transport) and indirect STT (with longer-range transport). I wonder whether this relates also to the STT process that is discussed in detail in the following study: Škerlak, B., Pfahl, S., Sprenger, M., and Wernli, H.: A numerical process study on*

*the rapid transport of stratospheric air down to the surface over western North America and the Tibetan Plateau, Atmos. Chem. Phys., 19, 6535–6549, https://doi.org/10.5194/acp-19 6535-2019, 2019.*

*At least, it seems to me that the important distinction between direct and indirect STT is also addressed in this numerical study. If so, the authors might want to refer to this study; if not, of course, there is no need to include it. I think it would be worthwhile to include some additional studies regarding direct/(indirect STT, as the discussion in L202-215 is certainly very plausible, but it is also somewhat more speculative than other parts of the paper]*

**Response 8:** Thank you for the introduction. Yes, a distinction between direct and indirect STT related to the STT process is discussed in detail in the study of Škerlak et al. (2019), we have included some additional studies regarding direct/indirect STT in the revised manuscript (lines 219–230) with the following discussions:

"Ozone STT events often involve the interaction of tropopause folds, long-range transport, and turbulent dynamics, spanning spatial scales of over 1000 km for transport processes and approximately 100 km for tropopause folds. The transport of stratospheric ozone in vertical layers, with transit times ranging from longer residence times (e.g., > 10 days) from thousands of kilometers away, to shorter times (lasting less than 2 days) in the troposphere can result from slow indirect STT and rapid direct STT ozone intrusions into the lower troposphere (Zanis et al., 1999, Eisele et al., 1999), and the important distinction of ozone intrusions between direct and indirect STT is also addressed in the numerical study by Škerlak et al. (2019). The impacts of direct intrusions of stratospheric ozone on central and eastern China are more pronounced at daily timescales and confined to limited areas, whereas indirect intrusions of stratospheric ozone originate from remote regions thousands of kilometers away, evenly distributed latitudinally along the westerlies between 40° and 70°N (Meng et al., 2024). Considering the difference in the spatial extent of the key regions with positive STT anomalies over China and surroundings between the first and second SVD modes…"

**References:**

Zanis, P., Schuepbach, E., Gaeggeler, H. W., Huebener, S., and Tobler, L.: Factors Controlling Beryllium-7 at Jungfraujoch in Switzerland. Tellus B 1999, 51 (4), 789–805. https://doi.org/10.1034/j.1600-0889.1999.t01-3-00004.x.

Eisele, H.; Scheel, H. E.; Sladkovic, R., and Trickl, T.: High-Resolution Lidar Measurements of Stratosphere–Troposphere Exchange. J. Atmospheric Sci. 1999, 56 (2), 319–330. https://doi.org/10.1175/1520-0469(1999)056<0319:HRLMOS>2.0.CO;2

Škerlak, B., Pfahl, S., Sprenger, M., and Wernli, H.: A numerical process study on the rapid transport of stratospheric air down to the surface over western North America and the Tibetan Plateau, Atmos. Chem. Phys., 19, 6535–6549, https://doi.org/10.5194/acp-19 6535-2019, 2019.

Meng, K., Zhao, T., Bai, Y., Wu, M., Cao, L., Hou, X., Luo, Y., and Jiang, Y.: Tracing the origins of stratospheric ozone intrusions: direct vs. indirect pathways and their impacts on Central and Eastern China in spring–summer 2019, Atmos. Chem. Phys., 24, 12623–12642, https://doi.org/10.5194/acp-24-12623-2024, 2024.

9. *[The analysis in Section 3.4 about meteorological mechanism is very nice, and clearly addresses some of my earlier concerns about cause-and-effect vs. simple correlation. Possibly, the authors can earlier in the manuscript already point to this section where good reasons for a cause-effect relationship is made.]*

**Response 9:** Thanks for the encouraging comments. Following the reviewer's suggestion, we have revised the last sentence of Sec. 3.2 (lines 254–255) as below:

"explores the meteorological mechanism of the cause-effect relationship in the following sections."

---

## Author Comment (AC2)

Dear Editors and Referees:

Thank you very much for your careful review and constructive comments on our manuscript acp-2025-737. We have accordingly revised the manuscript carefully. The revised portions are highlighted in the revised manuscript. In the followings, we quoted each review question in the square brackets and added our response after each question.

**Responses to Reviewer #2**

*[This work analyzes the role of Tibetan Plateau thermal forcing in stratosphere-troposphere transport through the Asian monsoon anticyclone and the westerly jet. Overall, the manuscript is very well written, the logic is sound, and the figures are clear. I did not find any major issues, and I recommend accepting it after minor revision.]*

**Response:** Many thanks for the encouraging comments and helpful suggestions on our manuscript. Following the reviewer's suggestions and comments, we have accordingly made careful revisions. Please find our point-to-point responses as follows:

*Major comments:*

1. *[Lines 44-59: some necessary introduction of the background dynamics was missing in this paragraph. For example, which altitude is the westerly, and which altitude of the two branches? How are they connected with the Asian monsoon anticyclone?]*

**Response 1:** Thank you for the suggestion. In the revised manuscript (lines 50–57), we have accordingly added some necessary introductions of the background dynamics as follows:

"Climatologically, the westerly jet is located near the tropopause between 200 hPa and 300 hPa, with the southern branch extending northeastward from the southern side of the Tibetan Plateau (TP) to the Northwest Pacific and the northern branch stretching southeastward from the 40°–65°N region north of the TP toward the Northwest Pacific

(Cressman, 1981). During spring and summer, the thermal forcing of the TP induces the Asian summer monsoon anticyclonic circulation in the upper troposphere and lower stratosphere (UTLS) (Wu et al., 2007), intensifying the northern branch westerly jet northward, and inhibiting the southern branch westerly jet, which may even disappear by late June (Duan and Wu, 2005; Luo and Yanai, 1984)."

**References:**

Cressman, G. P. (1981). Circulations of the West Pacific jet stream. Monthly Weather Review, 109(12), 2450-2463.

Wu, G., Liu, Y., Zhang, Q., Duan, A., Wang, T., Wan, R., Liu, X., Li, W., Wang, Z., and Liang, X.: The influence of mechanical and thermal forcing by the Tibetan Plateau on Asian climate, Journal of Hydrometeorology, 8, 770–789, 2007.

Duan, A., & Wu, G. (2005). Role of the Tibetan Plateau Thermal Forcing in the Summer Climate Patterns over Subtropical Asia. Climate Dynamics, 24, 793–807.

Luo, H., & Yanai, M. (1984). The Large-Scale Circulations and Heat Sources over the Tibetan Plateau and Surrounding Areas during the Early Summer of 1979. Part I: Precipitation and Kinematic Fields. Monthly Weather Review, 112(6), 966–989.

2. *[Section 2.2: TP covers a smaller region compared with the Asian monsoon region. I understand that the authors want to emphasize the role of the TP, however, figure 3 and 4a show some significant region near the border of the TP region as selected. Some discussion is necessary for the regions beyond the selected region.]*

**Response 2:** Thanks for your suggestion. In the revised manuscript (lines 242–249), we have accordingly added the dissuasions as follows:

"Although TP covers a smaller region compared to the broader Asian monsoon region, the thermal forcing over the TP, as "the world roof", is the decisive factor in the formation of the Asian monsoon anticyclone and plays a significant role in building the Asian monsoons (Wu et al., 2015; Duan et al, 2024; Xu et al., 2010). Fig. 3a and Fig. 4a encompass the region from 78°–103°E and 28°–38°N, covering most areas with elevations exceeding 3000 m, which are identified as the extent of the TP (Xu et al.,

2016). Moreover, the areas with heterogeneous correlations passing the significance test are primarily concentrated over the TP region including the TP platform within the 3000 m elevation border (black line) and some regions of the TP slopes near the TP platform. Therefore, our study focuses on this region to investigate the relationship between TP thermal forcing and STT variations."

**References:**

Wu, G., Duan, A., Liu, Y., Mao, J., Ren, R., Bao, Q., ... & Hu, W. (2015). Tibetan Plateau climate dynamics: recent research progress and outlook. National Science Review, 2(1), 100-116.

Duan, A., Wu, G., Wang, B., Turner, A. G., Hu, J., Hu, W., ... & Tang, Y. (2024). Drivers of East Asian summer monsoon variability: Global oceans and the Tibetan Plateau. Science Bulletin, 69(16), 2487-2490.

Xu, X., Lu, C., Shi, X., and Ding, Y.: Large-scale topography of China: A factor for the seasonal progression of the Meiyu rainband?, Journal of Geophysical Research: Atmospheres, 115, https://doi.org/10.1029/2009JD012444, 2010.

Xu, X., Zhao, T., Liu, F., Gong, S. L., Kristovich, D., Lu, C., Guo, Y., Cheng, X., Wang, Y., and Ding, G.: Climate modulation of the Tibetan Plateau on haze in China, Atmospheric Chemistry and Physics, 16, 1365–1375, https://doi.org/10.5194/acp-16-1365-2016, 2016.

*Specific comments:*

*1. [line 36: multiscale climate patterns: what are the patterns?]*

**Response 1:** Thanks for your comment. We have revised the sentence to enhance clarity in the revised manuscript (lines 36–38) as follows:

"tropospheric ozone can influence global radiation balance and then the multiscale climate patterns of global, hemispheric, and regional circulations in the atmosphere such as high-latitude warming, Hadley circulation, and East Asian summer monsoon system (Chen et al., 2007; Li et al., 2018)"

*2. [Line 126: 'over the Asia region' : to 'over Asia']*

**Response 2:** It has been corrected in the revised manuscript.

3. *[Line 138: 'the low period of air mass STT' to 'the period of low air mass STT']*

**Response 3:** It has been corrected in the revised manuscript.

4. *[Figure 3 and figure 4a: Is the black lines province boundary? This could be misleading and looks like contour of certain variables. I suggest use altitude in this figure since the focus of the figures are over the Tibetan Plateau.]*

**Response 4:** The black lines in Figs. 3a and 4a represent the 3000 m elevation contour outlining the area of the Tibetan Plateau.

5. *[Lines 328-332: this sentence is too long, please consider breaking into shorter sentences.]*

**Response 5:** Following the reviewer's suggestion, we have broken into the shorter sentences in the revised manuscript (lines 356–361) as follows:

"Through the SVD diagnostic analysis, we identify a synergistic relationship between interannual variations in the TP thermal forcing and the STT frequency. The intensifying thermal forcing in the central to eastern TP is closely linked to increasing and decreasing STT frequency respectively along the northern and southern branches of westerly jet, while the enhancing thermal forcing over the western TP is related to the western enhancement and eastern decline STT over China and surrounding regions. The thermal forcing changes over the TP regions could regulate the interannual STT anomalies over China by altering the direct and indirect pathways of ozone STT. "

---

## Author Response (AR2)

Dear Editor:

Thank you very much for your careful review comments on our manuscript EGUSPHERE-2025-737. In the following, we quoted the review question in the square brackets and accordingly presented our response whereafter.

**Responses to Editor**

*[I have one suggestion: given it was identified by one of the reviewers as unclear, please consider adding the description of what the black line in Fig. 3a and 4a indicates to the figure captions.]*

**Response:** Thank you for the careful review. We have accordingly added the description of what the black line in Fig. 3a and 4a indicates to the figure captions as follows:

"**Figure 3: Heterogeneous correlations of (a) TP thermal forcing, where the black line represents the isoline of altitude at 3000 m outlining the area of the TP, (b) STT frequency, where the red and blue dashed lines denote the northern and southern branches of STT frequency, respectively, and (c) their normalized temporal coefficients in the first SVD mode (SVD1). Black dots mean passing the 90 % confidence level based on the Student's t-test.**

**Figure 4: Same as Fig. 3, but for the second SVD mode (SVD2).**"